# Salvage Hepatectomy for Giant GIST Liver Metastases Unresponsive to Systemic Therapy—Case Report

**DOI:** 10.3390/life13081681

**Published:** 2023-08-03

**Authors:** Alin Kraft, Cosmin Moldovan, Alexandru Bârcu, Radu Dumitru, Adina Croitoru, Vlad Herlea, Irinel Popescu, Florin Botea

**Affiliations:** 1Department of General Surgery, “Regina Maria” Military Emergency Hospital, 500007 Brașov, Romania; alin.kraft@gmail.com; 2Department of General Surgery, Witting Clinical Hospital, 010243 Bucharest, Romania; 3Department of Medical-Clinical Disciplines, Faculty of Medicine, “Titu Maiorescu” University of Bucharest, 031593 Bucharest, Romania; adina.croitoru09@yahoo.com (A.C.); herlea2002@yahoo.com (V.H.); irinel.popescu220@gmail.com (I.P.); boteaflorin@yahoo.com (F.B.); 4Doctoral School in Medicine, “Titu Maiorescu” University, 040441 Bucharest, Romania; alexbarcu@gmail.com; 5Department of Radiology and Medical Imaging, “Fundeni” Clinical Institute, 050474 Bucharest, Romania; radudumitru@gmail.com; 6Department of Oncology, “Fundeni” Clinical Institute, 022328 Bucharest, Romania; 7Department of Pathology, “Fundeni” Clinical Institute, 022328 Bucharest, Romania; 8“Dan Setlacec” Center for General Surgery and Liver Transplant, “Fundeni” Clinical Institute, 022328 Bucharest, Romania

**Keywords:** liver resection, debulking, gastrointestinal stromal tumor, liver metastases, overall survival

## Abstract

Therapeutic decision-making for advanced GIST liver metastases is challenging due to limited clinical evidence. This case study aims to demonstrate the survival benefit of resection in non-responsive cases. A 40-year-old male presented with abdominal pain, weight loss, altered general status, massive hepatomegaly, and intermittent melaena. He was diagnosed with stage IV GIST with the primary tumor in the ileal loop and multiple gigantic synchronous bilobar liver metastases. Despite 31 months of tyrosine-kinase inhibitor therapy post-primary tumor resection, the disease remained unresponsive. The patient was admitted to our tertiary center with significant hepatomegaly. A two-stage debulking liver resection was performed after a multidisciplinary team decision. The first operation debulked the left hemiliver through a non-anatomical ultrasound-guided resection of segments 2, 3, and 4. The second operation (7 weeks later) debulked the right hemiliver through a right posterior sectionectomy involving segments 5 and 8. Despite receiving a second line of tyrosine-kinase inhibitor therapy after surgery, the disease progressed both within and outside the liver. However, the patient survived for 55 months, with a postoperative survival benefit of 25 months. In conclusion, this case emphasizes the significant survival benefit achieved through a complex two-stage debulking liver resection for giant liver metastases, even in cases where systemic therapy fails.

## 1. Introduction

Gastrointestinal stromal tumors (GISTs) are considered the most common type of mesenchymal tumors, with a reported annual estimated incidence of 3000 to 5000 cases in the United States [1]. In recent years, GIST liver metastases (GLM) have gained scientific interest due to their high incidence rates (23–47%) [2], frequent synchronous presentation (15–20%) [3], and specific liver affinity [4]. Nowadays, first-line TKI therapy can achieve disease control, increasing the median overall survival of GLM to 5 years [5]. However, patients present high reported rates of secondary mutation development, which are responsible for TKI therapy resistance [6]. In this setting, the role of liver resection as part of a multimodal therapeutic strategy was established by improving the overall survival of GLM patients [3]. Recently, publications in the literature have reported controversial data concerning liver resection as part of a multimodal therapeutic strategy in terms of the efficacy of debulking procedures in patients with progressive disease [7,8], as well as proper timing [9]. The current paper analyses the efficacy of a debulking liver resection in such a case (a patient presenting with giant GLM).

## 2. Case Presentation

We present the case of a 40-year-old Caucasian man (urban environment, tertiary education) with no relevant medical history (depression, hypertension, no previous abdominal surgery). He was admitted to the primary care hospital in July 2018, presenting with diffuse abdominal pain, significant weight loss (16 kg in the last 12 months), altered general status, hepatomegaly, and intermittent melaena (with associated anemia; Hb = 6.7 g/L). An abdominal MRI, a thoracic CT scan, and a percutaneous liver biopsy led to the diagnoses of a stage IV gastrointestinal stromal primary tumor located on an ileal loop (12 cm in size) and multiple large bilobar liver metastases ranging from 1 to 12 cm in diameter. The histological exam showed metastases from a gastrointestinal stromal tumor of enteral origin; mitotic count: 6/30 HPF, with CD 117 and CD 34 positive, cytokeratin AE1/AE3 and S100 negative, and a Ki-67 Index of 25% (Figure 1).

At this time, the treatment goal (set by a multidisciplinary team) was palliative because of the great disease burden in the liver, which was considered unresectable, and also because the ileal tumor did not show any acute complications (Hb = 9.3 g/L after administering 1 MER). Therefore, TKI therapy was initiated (Imatinib, 400 mg daily). The oncological follow-up at 2 months of TKI therapy showed a partial response according to the Choi criteria (the primary tumor measured 5.6 cm, and the liver metastases were stationary). Five months later, the primary tumor began to bleed significantly; therefore, emergency surgery involving a segmental ileal enterectomy with end-to-end anastomosis was performed. The preoperative MRI revealed stable disease according to the Choi criteria (Figure 2). The histological exam revealed an ileal gastrointestinal stromal tumor 6 cm in size, with a mitotic index greater than 5/50 HPF, with positive CD 117 and a Ki-67 Index of 25%, staging pT3 pN1 pM1hep (Figure 3).

One month after surgery, TKI therapy was reinitiated with Imatinib (400 mg/day). The oncological follow-up performed 3 months later via MRI showed progressive disease according to the Choi criteria (increase in the size of the liver metastases). Therefore, the Imatinib dose was doubled to 800 mg/day, and this dosage was administered for the next 4 months. Over the next 2 months, MRI imaging showed a further increase in size. Therefore, Sunitinib therapy (50 mg daily) was administered for the next 17 months, and this was well tolerated. Despite this treatment, the liver metastases slowly and constantly grew in size, with the largest measuring 176 mm at this point. In addition, their aspect became bulky, with a heterogeneous structure marked by hemorrhage and necrosis and accompanied by a right pleural effusion.

The patient was referred to our tertiary hepato-bilio-pancreatic surgery center to reassess GLM resectability. At this point, the patient was handicapped due to the giant metastases impeding walking and breathing; the ECOG performance status was 4, and the Charlson comorbidity index was 6. At imaging, aside from several small metastases in the central part of the liver (liver segments 4, 5, and 8), there were two large groups of confluence metastases: One was 17/10 cm in size, completely occupying segments 6 and 7, partly occupying segments 5 and 8, and invading the right hepatic vein. The other was 14/11 cm in size, occupying the entirety of segment 2, partly occupying segments 3 and 4 superior, and invading the left hepatic vein (Figure 4). Total liver volume was 5092 cm^3^; left hemiliver volume was 1876 cm^3^, while right hemiliver volume was 3217 cm^3^.

The case was discussed in a multidisciplinary setting, and debulking liver resection involving the resection of the two large groups of metastases in a two-stage approach was recommended. The first-stage surgery removed the left metastatic group by means of an echo-guided non-anatomical resection of segment 2, extended to segment 3 and 4 superior, along with the invaded left hepatic vein (Figure 5). The operative time was 240 min, and blood loss was 750 mL, necessitating 1 MER transfusion. At histologic examination, GIST liver metastases were confirmed, with a mitotic count of less than 5/50 HPF and a Ki-67 Index of 10%, DOG1 positive, and CD117 positive. Postoperative transient mild liver insufficiency was encountered on postoperative day (POD) 4 and lasted until POD 12 (maximum total bilirubinemia = 3.7 mg/dl, INR = 1.74)—Grade A according to ISGLS criteria, elevated cytolysis (AST = 298, ALT = 528), and ascites (maximum 1000 mL/day), with no clinical alterations. In addition, transient fever and leukocytosis was recorded on PODs 10 to 17. The patient was discharged on POD 24 in good condition.

Seven weeks after the first resection, we performed the second stage of the debulking procedure, which consisted of an intraoperative ultrasound-guided right posterior sectionectomy extended to segments 5 and 8, along with the resection of the invaded right hepatic vein, removing the large confluenced metastases located in the right hemiliver (Figure 6) with a portocaval shunt using Gore-Tex graft interposition to decrease post-resectional portal hypertension.

The operative time was 480 min, and blood loss was 2700 mL, necessitating a five-unit blood transfusion. The postoperative course was relatively uneventful, with the exception of fever on POD 11 (remitted by antibiotherapy) and right pleural effusion requiring drainage on POD 12. The patient was discharged on POD 19. The CT scan performed before discharge showed few remaining small liver metastases (the largest had a diameter of 26 mm) and asymptomatic small fluid accumulation on the liver cut surface (treated conservatively) (Figure 7).

After discharge, the multidisciplinary team decided to reinitiate the second line of tyrosine kinase inhibitor therapy. We had to employ the only reimbursed therapeutic alternative available in our country at that moment, Imatinib (400 mg daily), for the next 4 months, with progressive disease according to the Choi criteria. Consequently, the Imatinib dose was doubled to 800 mg/day. Five months later, a PET-CT exam revealed the following progressive disease: newly diagnosed size increase in the hepatic lesions and bone metastases located in the right iliac tuberosity, vertebras (C3, T3, L5, and right humerus). Therefore, for the next 4 months, the patient was subjected to therapy involving 50 mg of Sunitinib daily and 300 mg of tramadol daily. The disease progressed in the liver (new liver metastases with a further increase in size) and bones (increase in both size and number of the bone metastases located in both iliac wings, the right sacrum wing, right iliac tuberosity, vertebras -C3, C7, T3, L2, L5-S1, and right humerus). One month later, after a total time length of 46 months from the initial diagnosis, the patient suffered severe depression, becoming uncompliant to therapy and ceased the oncological follow-up. The patient was admitted 9 months later into the emergency ward for seizures; the cerebral MRI showed skull metastases with loco-regional intra- and extracranial invasion. The patient was discharged upon request and died shortly after. The overall survival was 55 months from diagnosis, with a postoperative survival benefit of 25 months.

## 3. Discussion

The case described in this paper illustrates the overall survival benefits of the complex and aggressive stand-alone surgical management of advanced-stage GLM when faced with poor response to TKI therapy. Given that the presentation was considered unresectable even in the earlier stage, the specificity of the case necessitated ensuring the resectability of GLM by using a two-stage approach, which was carried out by a team with expertise in liver surgery. Consequently, we recommend that debulking should always be considered in such cases, and resectability has to be assessed by surgeons experienced in liver resection.

In the current era, which has been defined by the successful introduction of TKI therapy for metastatic GIST treatment, the data in the literature data state that over 80% of patients initially show disease control to first-line therapy [5]. However, complete response rates are low (≥6%) [10], and most patients show a median progression-free survival of just 24 months [11]. Therefore, surgical management has taken on a new role in the multimodal therapeutic approach for metastatic GISTs. It is generally thought that surgery delays disease progression and prolongs survival in selected patients by potentially removing resistant clones, reducing tumor burden and preventing the development of secondary mutations [5,10,12,13].

Recently, controversial data regarding the benefit of liver resection as part of a multimodal therapeutic strategy have been reported [14]. Some authors suggest that only patients who present isolated sites of progression, partial response, or stable disease post-TKI therapy are likely to have improved survival following surgical management [15]. Although resection is generally considered unhelpful for patients who present disease progression during TKI therapy [3], several authors have reported improved progression-free and overall survival in patients with progressive disease undergoing a complete resection of all metastatic sites [8]. Nevertheless, some literature data promote the concept of debulking liver resection in such cases to remove TKI-resistant clones, thus achieving improved progression-free survival if the remanent disease is TKI-responsive [16,17], or to achieve symptomatic control (bleeding, pain, or bowel obstruction) [18]. However, the post-resectional overall survival benefit of 25 months encountered in adverse conditions (absence of TKI responsiveness and psychiatric-related causes of treatment non-compliance) reported in the current paper provides evidence of survival benefit despite high-liver disease burden, which would be considered unresectable.

The existing literature promotes the use of nonanatomical parenchyma sparring resections, especially for small peripheral lesions located away from major vascular structures, because of the well-circumscribed characteristics of GLM [3,19]. We did not deem it feasible to perform single-stage liver debulking due to the insufficient future liver remanent volume and technical difficulties (tumoral adherence to the diaphragm and intensive collateral venous circulation); therefore, we decided to perform two-stage liver debulking. The postoperative course proved that the strategy was the appropriate course of action, and we recommend it in cases such as the one presented in this paper. Based on the overall uneventful postoperative period, TKI therapy was rapidly reinstituted.

The optimal timing of liver resection continues to be a subject of debate, although many authors agree with administering TKI therapy for at least 6 months after and subsequently considering surgical management depending on the disease response [20,21]. In our case, the follow-up performed after the initial introduction of first-line TKI therapy reported a partial response. This timeframe is considered by several authors to be the opportune time to perform surgical management [20]. However, the primary tumor began to bleed significantly; therefore, emergency resection was performed. Given the emergency setting, it was inappropriate to perform a synchronous technically demanding liver resection. We believe that the current case would have encountered a different outcome if the liver debulking resection had been performed at that time, resulting in low remanent tumor burden and a lower risk of developing secondary mutations.

The current guidelines regarding patients with metastatic GIST state that the best response rates are achieved when TKI therapy initiation immediately follows the diagnosis [20]. If liver resection is feasible, TKI therapy must be administered up to the date of surgery, restarted early postoperatively, and continued indefinitely, even if faced with complete resection [22]. If disease progression occurs, the cessation of TKI therapy would lead to shortened progression-free and overall survival rates [20].

The patient discussed in this paper was subjected to TKI therapy in accordance with the appropriate guidelines [22]. First-line TKI therapy was immediately introduced following diagnosis of the long-time neglected and advanced-stage disease. Following documented disease progression, we chose to double the dose of first-line TKI therapy (because the literature data reports improved disease response as a result [3]), and later on, second-line TKI therapy was introduced. This prevented the development of extrahepatic metastases but led to the slow and constant growth of the liver-located disease. Given the development of extrahepatic metastases, some may question the utility of the first-line TKI rechallenge therapy introduced immediately after liver resection, but at that time, it was the only reimbursed therapeutic option available nationwide; in addition, the discounted second-line TKI therapy was introduced as soon as possible. The follow-up revealed intra-, and extrahepatic disease progression, underlying the existence of secondary resistant mutations and rendering the patient psychically unable to participate in any type of therapy due to severe depression.

Upon assessing the treatment response based upon imagistic measurements, we decided to apply the Choi [23] vs. RECIST criteria [24], as literature shows that GIST surveillance requires evaluating both tumor density and size (i.e., tumor progression is often seen as new areas of hyper density before an increase in size) [25].

Publications in the literature describe both a primary and secondary resistance to TKI treatment. Often, tumors develop secondary resistance, resulting in first-line TKI therapy failure due to secondary KIT or PDGF-R mutations [26]. It is known that the presence or absence of mutations in specific regions of the KIT and PDGFRA genes are correlated with a response (or lack of response) to specific TKIs. Therefore, current guidelines strongly recommend determining the above-mentioned when TKIs are considered as part of the therapeutic plan. Unfortunately, the nationwide government-funded therapeutic plan does not include the determination of mutations. Therefore, due to the patient’s poor financial status, the tumoral mutations were impossible to characterize. Nevertheless, we believe that secondary resistance was easily developed based on the patient’s high tumor burden, which was present at the moment of initial diagnosis. In addition, second-line TKI therapy was immediately introduced in the presence of such advanced disease, in accordance with existing guidelines.

Reports regarding liver transplantation for GLM are scarce in the current literature [27,28]. It is thought that current liver transplantation expertise is applicable in a neoplastic setting only in managing colorectal and neuroendocrine tumors liver metastases [29]. Nonetheless, some studies consider this setting to be characterized by evidence paucity and propose selection criteria [29]. We did not consider liver transplantation a feasible therapeutic option due to the high risk of systemic recurrence, tumor aggressiveness (a high Ki-67 index of 25%), lack of TKI response, worldwide limited experience, and because post-transplant immunosuppressive therapy may favor tumor recurrence.

## 4. Conclusions

The case described in this paper illustrates the overall survival benefit that can be derived from a complex, technically demanding two-stage debulking liver resection performed in a high-volume tertiary hepato-biliopancreatic surgery center as part of a multimodal therapeutical approach when faced with advanced liver metastases unresponsive to TKI therapy.

## Figures and Tables

**Figure 1 life-13-01681-f001:**
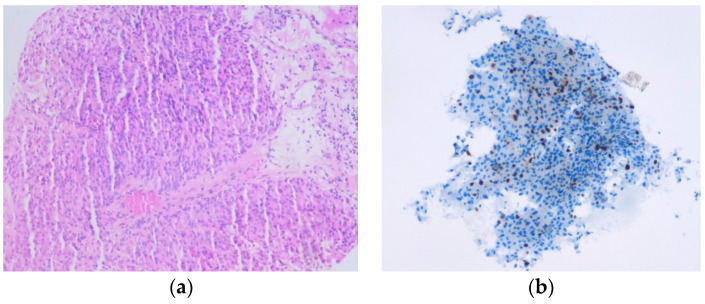
(**a**) Microscopic aspect magnification 50×—hematoxylin eosin staining showing liver metastases from a gastrointestinal stromal tumor of enteral origin; mitotic count: 6/30 HPF. (**b**) Microscopic aspect magnification 200×—immunohistochemistry staining showing liver metastases from a gastrointestinal stromal tumor of enteral origin: CD 117 and CD 34 positive, cytokeratin AE1/AE3 and S100 negative, and a Ki-67 Index of 25%.

**Figure 2 life-13-01681-f002:**
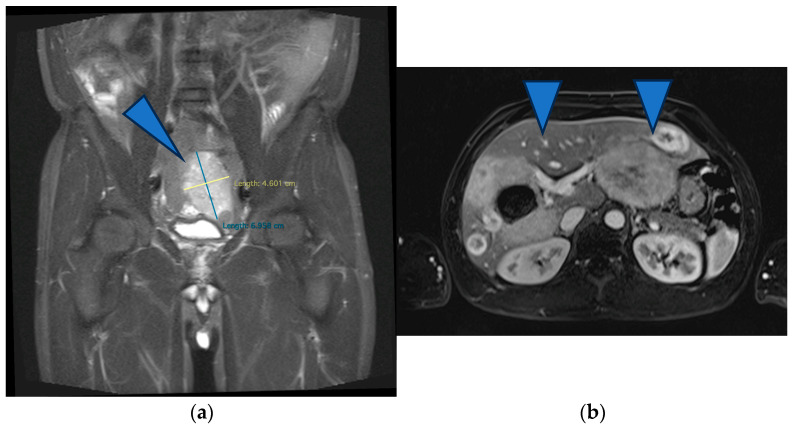
(**a**) MRI showing stable disease according to the Choi criteria of the primary tumor (arrowhead) (**b**) and of the liver metastases (arrows).

**Figure 3 life-13-01681-f003:**
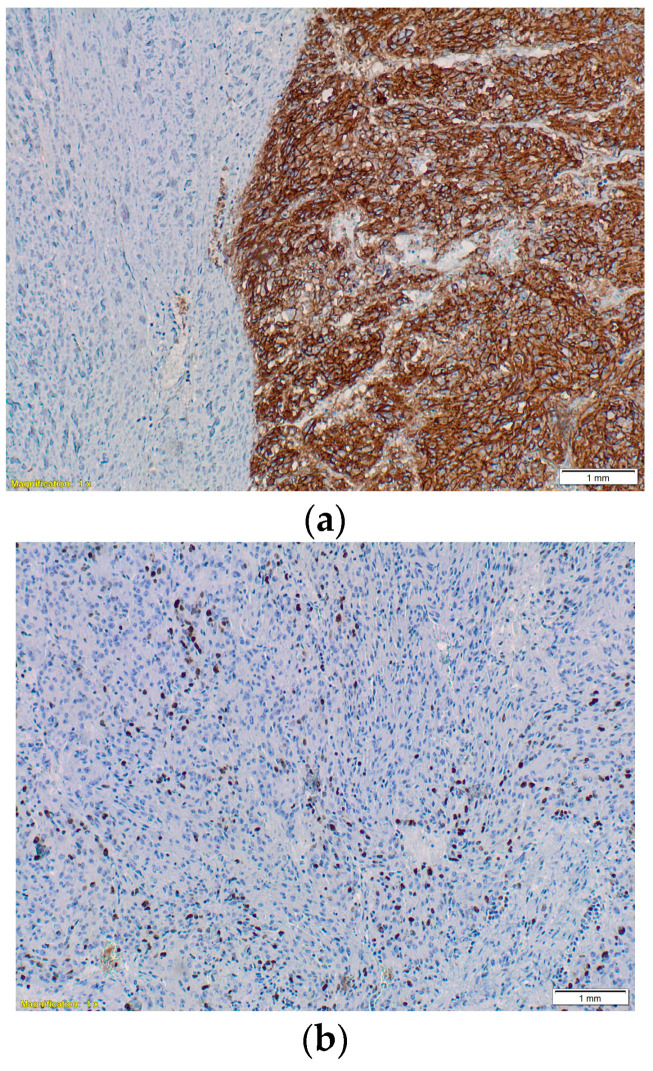
(**a**) Immunohistochemistry staining showing an ileal gastrointestinal stromal tumor CD 117 positive, microscopic aspect magnification 100×. (**b**) Microscopic aspect magnification 100× with a KI-67 index of 25%.

**Figure 4 life-13-01681-f004:**
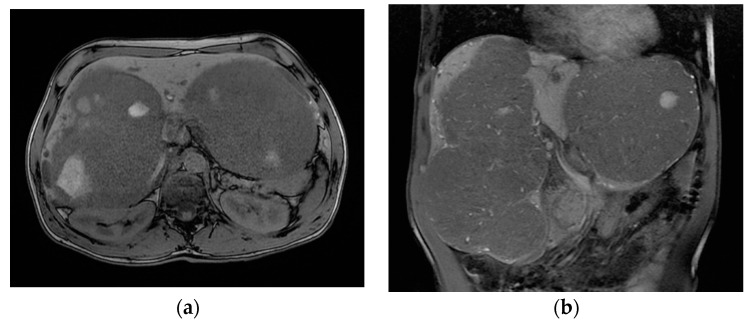
MRI showing the two groups of large liver metastases: (**a**) one group located in segments 6 and 7 and partly in segments 5 and 8, invading the right hepatic vein, while (**b**) the other occupied the entirety of segment 2 and partly occupied segments 3 and 4 superior, invading the left hepatic vein.

**Figure 5 life-13-01681-f005:**
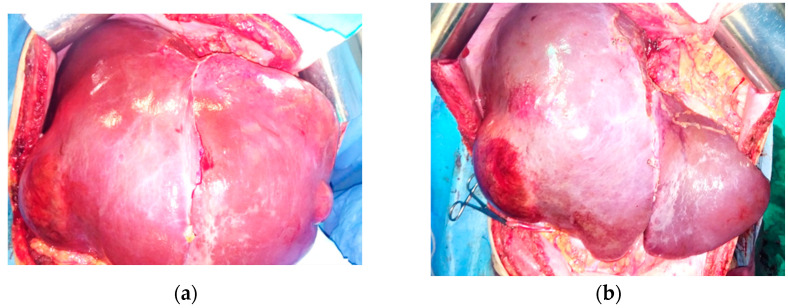
First-stage surgery. Aspect of the liver: (**a**) prior to first-stage resection; (**b**,**c**) after resection; (**d**) resected specimen.

**Figure 6 life-13-01681-f006:**
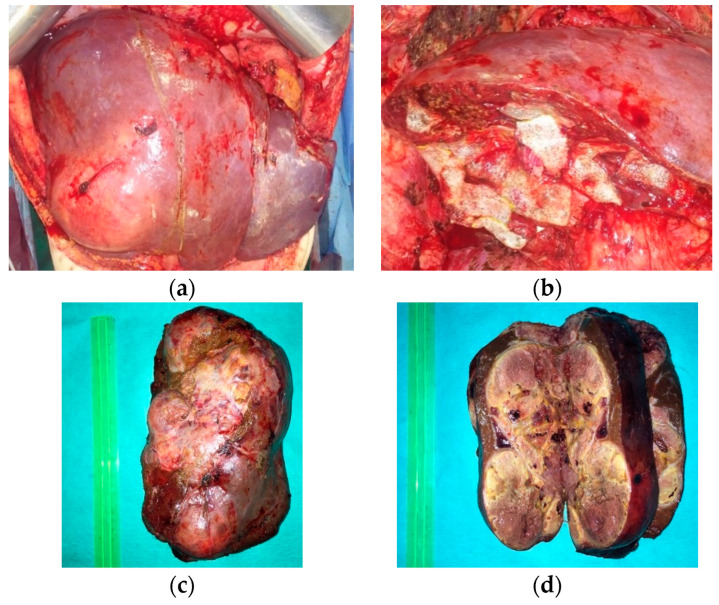
Second-stage surgery. Aspect of the liver: (**a**) prior to second resection; (**b**) after resection; (**c**,**d**) resected specimen.

**Figure 7 life-13-01681-f007:**
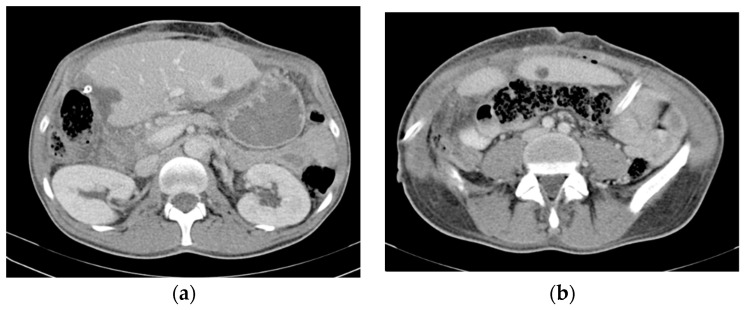
Contrast-enhanced postresection CT image showing (**a**) a few remaining small liver metastases (the largest of which had a diameter of 26/17 mm) (**b**) and an asymptomatic collection on the liver resection surface.

## Data Availability

All clinical and imaging data can be obtained upon written request from the corresponding author.

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
