# Peer review of "Salvage Hepatectomy for Giant GIST Liver Metastases Unresponsive to Systemic Therapy—Case Report"

_life, 2023, doi:10.3390/life13081681_

Round 1

Reviewer 1 Report

The authors described a patient of stage IV GIST with massive liver metastasis. The disease progressed despite primary tumor resection and TKI therapy. Refractories to first-line and second-line TKI was noted. A two-stage debulking liver resection was performed. The patient survived for 55 months. The authors concluded the significant survival benefit achieved through a complex two-stage debulking liver resection for giant liver metastases, even in cases where systemic therapy fails. This case is interesting, but the manuscript needs much to be improved.

1. The Introduction section is poor written. The sentences are not very coherent, and there are too many paragraph breaks.
2. In the Case Presentation section, the narrative is excessively lengthy and should be condensed. 

3. According to the guideline, testing for KIT and PDGFRA mutations should be performed if TKIs are considered as part of the treatment plan since the presence of mutations (or absence of mutations) in specific regions of the KIT and PDGFRA genes are correlated with response (or lack of a response) to specific TKIs. However, I didn't see the author performing tumor genetic analysis for the patient. 

4. Figure 1 is not legible and there are too many redundant images, such as CD117 immunostain. 

I suggest sending the English editing before submitting your manuscript.

Author Response

  1. The Introduction section is poor written. The sentences are not very coherent, and there are too many paragraph breaks.

Response: We modified the Introduction section accordingly, we re-read the text, and improved the flaws language wise (rows: 42-59).

  1. In the Case Presentation section, the narrative is excessively lengthy and should be condensed.

Response: We modified the Case Presentation section accordingly, we re-read the text, and improved the flaws language wise (rows: 61-447).

  1. According to the guideline, testing for KIT and PDGFRA mutations should be performed if TKIs are considered as part of the treatment plan since the presence of mutations (or absence of mutations) in specific regions of the KIT and PDGFRA genes are correlated with response (or lack of a response) to specific TKIs. However, I didn't see the author performing tumor genetic analysis for the patient. 

Response: “It is known that the presence or absence of mutations in specific regions of the KIT and PDGFRA genes are correlated with response (or lack of response) to specific TKIs. Therefore, current guidelines strongly recommend to determine the abovementioned when TKIs are considered as part of the therapeutic plan. Unfortunately, the nation-wide government funded therapeutic plan does not include the mutation determining. Therefore, due to the patient’s poor financial status, the tumoral mutations were impossible to characterise”. We added the paragraph to the discussions section and added the required bibliographic citations (rows: 714-720).

  1. Figure 1 is not legible and there are too many redundant images, such as CD117 immunostain. 

Response: We modified the manuscript accordingly. We eliminated the redundant images.

  1. I suggest sending the English editing before submitting your manuscript.

Response: We re-read the whole manuscript and improved the flaws language wise. The language editing work was provided by professional assistance.

Reviewer 2 Report

In this case report, the he overall survival benefit brought by a complex techni- 289 cally demanding 2-stage debulking liver resection in a stage 4 GIST patient with resistance to TKI therapy is described.

The case report is well presented, and the rationale and current state of the art treatment algorithms are discussed. Minor changes to the manuscirpt should be undertaken:

Was genotyping performed in this case? What were the mutations revealed by histopathology, as explanation for the TKI resistance and rationale for the dose increment of TKI therapy?

The treatment approach of cytoreductive surgery in TKI resistant stage 4 GIST is not novel, although the postoperative survival benefit is remarkable.

Author Response

  1. Was genotyping performed in this case? What were the mutations revealed by histopathology, as explanation for the TKI resistance and rationale for the dose increment of TKI therapy?

Response: “It is known that the presence or absence of mutations in specific regions of the KIT and PDGFRA genes are correlated with response (or lack of response) to specific TKIs. Therefore, current guidelines strongly recommend to determine the abovementioned when TKIs are considered as part of the therapeutic plan. Unfortunately, the nation-wide government funded therapeutic plan does not include the mutation determining. Therefore, due to the patient’s poor financial status, the tumoral mutations were impossible to characterise”. We added the paragraph to the discussions section and added the required bibliographic citations (rows: 714-720).

“We chose to increase the dose of TKI therapy, because literature suggests performing this first step to improve disease response”. We added the paragraph to the discussions section and also added the required bibliographic citations (rows: 696-698).

  1. The treatment approach of cytoreductive surgery in TKI resistant stage 4 GIST is not novel, although the postoperative survival benefit is remarkable.

Response: “We consider that the current case particularity consists in insuring the resectability of GLM by using a two stage approach performed by a team with expertise in liver surgery, in a presentation considered unresectable even in early stage. Consequently, we recommend that debulking should always be considered in such cases, and the resectability has to be assessed by experienced surgeons in liver resection.”. We added the paragraph to the discussions section (rows: 451-620).

Round 2

Reviewer 1 Report

I have no further comments.

I have no further comments.